# Migration Characteristics of Manure-Derived Antibiotic-Resistant Bacteria in Vegetables Under Different Soil Types

**DOI:** 10.3390/microorganisms13102398

**Published:** 2025-10-20

**Authors:** Tingting Song, Changxiong Zhu, Honghui Teng, Binxu Li, Shuang Zhong, Yan Qin, Jiawei He, Hongna Li

**Affiliations:** 1College of Engineering, Jilin Normal University, Siping 136000, China; songtingting0505@163.com (T.S.); tenghonghui@163.com (H.T.); ongzhish@126.com (S.Z.); qin.yan.19891017@163.com (Y.Q.); hejiaweispjl@163.com (J.H.); 2Institute of Environment and Sustainable Development in Agriculture, Chinese Academy of Agricultural Sciences, Beijing 100081, China; zhucx120@163.com (C.Z.); libinxu123@163.com (B.L.); 3College of Environmental Science and Engineering, Hebei University of Science and Technology, Shijiazhuang 050018, China

**Keywords:** pakchoi, soil type, CTC-manure application, chlortetracycline-resistant bacteria, microbial community structure

## Abstract

The application of livestock manure can introduce antibiotic-resistant bacteria (ARB) into soil–vegetable systems, potentially shaping the antibiotic resistance profiles of plants. This study investigated the effects of manure containing chlortetracycline (CTC) on antibiotic resistance in pakchoi grown in three distinct soil types (black soil, fluvo-aquic soil, and red soil). The results demonstrated that CTC-manure application significantly increased the relative abundance of CTC-resistant endophytic bacteria (CREB), with the magnitude of the increase following the order: black soil (235.43%) > fluvo-aquic soil (64.5%) > red soil (10.68%). Furthermore, the presence of CTC promoted the proliferation of *Acinetobacter* and increased the abundance of potential pathogens (such as *Klebsiella*, *Rhodococcus*, and *Corynebacterium*), thereby elevating the risk of antibiotic resistance transmission. Metabolomic analysis revealed that CTC induced the most substantial metabolic alterations in pakchoi grown in red soil. Correlation analysis indicated that the CREB community structure was primarily shaped by soil properties, including total nitrogen, total phosphorus, and organic matter, and was significantly correlated with indigenous soil ARB (*Pseudomonas*, *Bacillus*, and *Stenotrophomonas*). This study elucidates the mechanisms underlying manure-driven antibiotic resistance dissemination in vegetable production systems and offers a theoretical foundation for developing agricultural practices to mitigate associated risks.

## 1. Introduction

Since penicillin’s discovery in the 20th century, antibiotics have been widely used in human medicine, as well as in livestock and aquaculture feed [1,2]. This broad use has led to a significant increase in antibiotic-resistant bacteria (ARB) and associated antibiotic resistance genes (ARGs) in the intestines of both humans and animals [3]. Organic fertilizers provide essential macro- and micronutrients, along with abundant organic matter, making them popular choices in organic agriculture and green food production [4,5]. The application of livestock manure containing these ARB as an organic fertilizer can lead to further dissemination and proliferation of antibiotic resistance, exerting profound impacts on soil ecosystems [6,7,8]. Livestock manure serves as a significant reservoir of ARB. Long-term application promotes the proliferation and dissemination of manure-derived ARB in soils, which may subsequently transfer to soil microbiota via horizontal gene transfer (HGT). Notably, certain indigenous microorganisms (e.g., bacteria, fungi) possess intrinsic antibiotic resistance—a trait that has been refined over millions of years of exposure to antibiotics. In microbial communities, these antibiotics function dually as competitive weapons and signaling molecules, mediating both intra- and interspecies interactions [9]. Furthermore, human activities facilitate the spread of antibiotic resistance in soil, and soil fauna serve as important carriers of ARB. These combined effects make soil both a significant source and sink for ARB [10]. Studies have demonstrated that soil samples amended with swine manure exhibited the most significant increase in populations of ARB, including *Pseudomonas*, *Escherichia*, *Providencia*, *Salmonella*, *Bacillus*, *Alcaligenes*, and *Paenalcaligenes* [11,12]. ARB can be transmitted to vegetable crops through soil and water systems following the application of manure-based fertilizer [13]. This transmission pathway contributes significantly to the widespread dissemination of antibiotic resistance in environmental settings.

Plants harbor a substantial microbial community, known as the plant microbiome, which plays a critical role in plant health, fitness, and productivity [14,15]. Extensive research has focused on the diversity and abundance of ARB in soil. However, studies examining their subsequent transfer into the plant microbiome remain limited [11,14,16]. Studies have shown several pathways for ARB transmission from soil–root systems to plants. Resistant microorganisms can enter plants through root wounds or via aerial parts from the air. Plants can acquire resistance through HGT from other endophytic bacteria. In addition, soil fauna promotes ARB entry into plants through roots during rhizosphere activities [17,18]. Fluorescence labeling experiments have confirmed that *Escherichia coli* carrying the RP4 resistance plasmid can invade *Arabidopsis thaliana* through the root system and move to aerial parts. *Escherichia coli* can also transfer to soil microbial communities via HGT and is subsequently acquired by plant endophytes [19]. A relatively high prevalence of multiple antibiotic-resistant endophytic bacteria was consistently detected in pakchoi. The abundance approached 1% in samples amended with chicken manure composting [20].

ARB may colonize plant tissues through rhizospheric interactions by establishing themselves as endophytic bacteria, thereby introducing exogenous antibiotic resistance into the plant endophytic system. This ecological process effectively transforms crop plants into functional reservoirs of ARB, ultimately facilitating their transmission to humans through food consumption [21,22]. The plant microbiome serves as a critical interface between human-associated microbial communities and natural environmental microbiomes [14,23]. Endophytic ARB deserve more focus than phyllosphere-resistant bacteria, as internalized microbes cannot be removed by standard produce washing [16,24]. Therefore, the impact of fertilizer application on plant endophytes may ultimately affect human health through more severe potential threats. Furthermore, plants can absorb antibiotics from manure-amended soils, exerting selection pressure on endophytic communities and further facilitating the transmission and dissemination of ARB to humans [22,25]. However, current research on the effects of soil types on endophytic ARB in plants remains scarce. Pakchoi (*Brassica chinensis*) is a staple leafy vegetable in many parts of the world, with particularly high cultivation and consumption levels throughout East Asia. Its safety directly impacts a significant portion of the vegetable supply chain [26]. Therefore, this study investigates the diversity of ARB in the endophytic systems of pakchoi grown in different soil types (black, fluvo-aquic and red soils) under various manure treatments, aiming to: (1) explore the abundance and community structure variations of ARB in pakchoi endophytic systems under different soil types and manure treatments; (2) reveal the differential effects of soil types on ARB colonization; and (3) elucidate the transmission mechanisms of ARB along the manure–soil–plant continuum. This research provides new scientific evidence for understanding the transmission mechanisms and ecological risks of ARB in manure–soil–plant systems.

## 2. Materials and Methods

### 2.1. Experimental Design

The specific experimental design is the same as that of a previous study [12]. The leaching experiment used columns made of unplasticized polyvinyl chloride (UPVC). Each column measured 45 cm in height and 25 cm in diameter. The soils loaded into the columns were allowed to activate for 2–3 days prior to the experiment. Pakchoi (*Brassica chinensis*) seeds were sown in the surface soil layer. Irrigation was performed on days 4, 8, 15, 22, and 29 using deionized water as the experimental water source to avoid introducing antibiotics and ARB. The total experimental duration was 35 days. Upon completion of the cultivation period, pakchoi were collected, roots were excised, and the edible portions were placed in sealed bags for transportation to the laboratory for subsequent processing. The physical and chemical properties and CTC of the soils were characterized in our previous study [12]. Experimental treatments included: black soil (S1), black soil with manure (S1M), and black soil with chlortetracycline (CTC)-manure (S1MA), fluvo-aquic soil (S2), fluvo-aquic soil with manure (S2M), and fluvo-aquic soil with CTC-manure (S2MA), red soil (S3), red soil with manure (S3M), and red soil with CTC-manure (S3MA).

### 2.2. Plant Sample Collection and Processing

First, the pakchoi samples were rinsed to remove surface dirt and impurities, followed by three thorough washes with deionized water. The samples were then surface-disinfected by immersion in a 3% hydrogen peroxide (H_2_O_2_) solution for 30 min. Afterward, they were rinsed three times with sterile deionized water and soaked in 70% ethanol for 1 min. Subsequently, another thorough rinsing with sterile deionized water was performed. After sterilization, the samples were blotted dry with sterile filter paper for further processing. To verify the effectiveness of the surface sterilization, 100 μL of the final sterile deionized water rinse was spread onto Luria–Bertani (LB) agar plates and incubated at 35 °C for 24 h. The absence of bacterial growth on the plates confirmed successful surface sterilization of the vegetable samples.

### 2.3. Determination of Total Culturable Endophytic Bacteria and ARB Abundance in Pakchoi

After surface sterilization, the edible portions of pakchoi were aseptically cut into small pieces using sterile scissors. A 5 g sample was accurately weighed and placed in an Erlenmeyer flask containing 45 mL of sterile phosphate-buffered saline (PBS, 0.01 mol/L). The mixture was then shaken at 200 rpm for 30 min using an orbital shaker. After shaking, 1 mL was transferred to 9 mL PBS buffer for serial dilution. For the determination of antibiotic-resistant bacteria (ARB), 100 μL of appropriate dilutions were spread-plated onto both plain LB agar and LB agar supplemented with 16 mg/L chlortetracycline (CTC), according to a previous study [27] that followed Clinical and Laboratory Standards Institute (CLSI) guidelines. All plates were prepared in triplicate and incubated at 35 °C for 24 h. Colony-forming units (CFUs) were then counted to quantify the total cultivable endophytic bacteria (TCEB) and chlortetracycline-resistant endophytic bacteria (CREB).

### 2.4. Pretreatment of TCEB and CREB

All colonies grown on plates after 24 h were collected and washed with ddH_2_O, followed by centrifugation. To minimize culture bias associated with differential growth rates in liquid medium, all collected colonies were directly transferred into 5 mL centrifuge tubes. The samples were thoroughly vortex-mixed and centrifuged at 10,000 rpm for 5 min, after which the supernatant was discarded. Genomic DNA was extracted using the TIANamp Bacteria DNA Kit (TIANGEN, Beijing, China). The quality of extracted DNA was assessed by 1% agarose gel electrophoresis, while DNA concentration was measured using a DeNovix DS-11 Spectrophotometer (DeNovix, Wilmington, DE, USA). DNA samples that passed quality control were subsequently used for high-throughput sequencing.

### 2.5. 16S rRNA High-Throughput Sequencing

The high-throughput sequencing was performed on the Illumina MiSeq PE300 platform at Shanghai Majorbio Bio-pharm Technology Co., Ltd. (Shanghai, China). The V3–V4 hypervariable regions of the bacterial 16S rRNA gene were amplified using the universal primers 338F (5′-ACTCCTACGGGAGCAGCAG-3′) and 806R (5′-GGACTACHVGGGTWTCTAAT-3′). The raw sequencing data underwent quality control and assembly. Subsequently, singletons (sequences appearing only once) were removed by dereplicating the sequences with Usearch. Operational taxonomic unit (OTU) clustering was then performed with Usearch, grouping sequences into distinct OTUs based on 97% sequence similarity threshold. Representative sequences from each OTU were taxonomically classified by the RDP classifier Bayesian algorithm against reference databases. This pipeline ultimately generated a biological observation matrix detailing OTU abundances and their taxonomic affiliations.

### 2.6. Data Analysis

Data visualization and statistical analyses were performed using the following software tools: column charts were generated using Origin 9.1 (OriginLab, San Diego, CA, USA); one-way analysis of variance (ANOVA) was conducted using IBM SPSS Statistics 23.0 (IBM, Chicago, IL, USA) to determine significant differences among treatments (*p* < 0.05); phylum level variations in bacterial communities were visualized using Circos-0.67-7 (http://circos.ca/, accessed on 8 July 2025); heatmaps were constructed using the PcoA (Principal Coordinate Analysis) package in R 3.5.2 (https://www.r-project.org/, accessed on 8 July 2025); network analysis was performed and visualized using Cytoscape 3.3.0 (https://cytoscape.org/, accessed on 8 July 2025).

## 3. Results

### 3.1. Changes in the Abundance of Chlortetracycline-Resistant Bacteria in Pakchoi

The TCEB abundance ranged from 5.37 × 10^5^ to 7.63 × 10^5^ CFU/g in control treatments. In contrast, the absolute abundances of TCEB in manure treatments (S1M, S2M and S3M) and CTC-manure treatments (S1MA, S2MA and S3MA) were 1.64 × 10^6^–2.08 × 10^6^ CFU/g and 1.50 × 10^6^–2.08 × 10^6^ CFU/g, respectively (Figure 1a), representing an increase of approximately one order of magnitude compared to the control. The manure treatments showed the highest relative abundance of TCEB, following the order S1M > S2M > S3M, with the abundances of 2.64 × 10^6^ CFU/g, 2.10 × 10^6^ CFU/g, and 1.72 × 10^6^ CFU/g, respectively. Compared to manure treatments, the TCEB absolute abundance exhibited a significant reduction in the CTC-manure treatments (*p* < 0.05), though the final abundance levels remained higher than those in the control treatments.

The absolute abundance of CREB was significantly lower in S1 (3.61 × 10^2^ CFU/g) than in S2 and S3 treatments (*p* < 0.05). The CTC-manure markedly increased the abundance of CREB in pakchoi by approximately one order of magnitude (1.71 × 10^3^–3.61 × 10^3^ CFU/g). Comparative analysis revealed differential distribution patterns of CREB within the endophytic microbiota across different soil types. CREB abundance followed the order S2M > S1M > S3M in manure treatments, while the order was S1MA > S2MA > S3MA in CTC-manure treatments. The highest absolute abundance of CREB was 3.61 × 10^3^ CFU/g in S1MA treatment, which was significantly greater than that in S1M treatment (*p* < 0.05). In contrast, no significant difference was observed between S2M and S2MA treatments (*p* > 0.05). The CTC-manure substantially enhanced the relative abundance of CREB, with the most pronounced effects observed in black and fluvo-aquic soils (Figure 1b). The relative abundance of CREB in the S1M treatment (6.91 × 10^−4^) showed no significant difference compared to S1 treatment, whereas it demonstrated a significant increase (6.91 × 10^−3^) in S1MA treatment. The relative abundance of CREB significantly increased in S2M and S2MA treatments compared to S2 treatment, following the order: S2MA (1.46 × 10^−3^) > S2M (1.18 × 10^−3^) > S2 (8.88 × 10^−4^). Among CTC-manure treatments, CREB relative abundance was manifested as S1MA > S2MA > S3MA.

### 3.2. Changes in the Community Structure of Chlortetracycline-Resistant Endophytic Bacteria

Proteobacteria was the most abundant phylum of CREB across all treatments, with relative abundances ranging from 50.16% to 100%. Notably, Bacteroidetes also accounted for a considerable proportion (49.02%), coexisting with Proteobacteria in the fluvo-aquic soil control treatment (S2) (Figure 2a). The codominance of these two phyla within the endophytic compartment underlines the possible synergistic or competitive interactions influencing the assembly of CREB in pakchoi plants grown in this specific soil type.

Analysis of the top 15 genus level CREB in the pakchoi endophytic system revealed that *unclassified_f_Enterobacteriaceae* and *Enterobacter* were consistently present across all treatments (Figure 2b). In black soil, the relative abundances of *Bacillus* and *Paenibacillus* increased in S1M treatment compared to S1 treatment. In contrast, *Pseudomonas*, *Serratia*, and *Acinetobacter* were the dominant CREB in S1MA treatment, with relative abundances of 69.32%, 13.04%, and 12.76%, respectively. In the endophytic system of pakchoi in fluvo-aquic soil, the relative abundance of *Stenotrophomonas* increased markedly in the S2M treatment (66.67%) compared to the control (16.62%). Other CREB such as *Glutamicibacter* and *Pseudomonas* also showed increased relative abundances in the S2MA treatment. A similar enrichment pattern was observed for *Pseudomonas* in the S3MA treatment. The relative abundance of *Pseudomonas* increased in CTC-manure treatments. *Erwinia* became the dominant CREB genus in S3MA treatment, accounting for 82.85%. The study revealed that *Erwinia* exhibited increased abundance exclusively in S3MA treatment.

### 3.3. Analysis of Differential Genera of Chlortetracycline-Resistant Bacteria Under Different Treatments

The composition of CREB communities was compared in pakchoi cultivated across three soil types. Minimal differences in bacterial genera composition were observed between black soil and red soil. Five bacterial genera, *Rhodococcus*, *Enterobacter*, *unclassified_f_Enterobacteriaceae*, *Pseudomonas*, and *Stenotrophomonas* were consistently present in all treatments of black soil (Figure 3a). An increasing trend in unique CTC-resistant bacteria (CRB) genera was observed in S1M and S1MA treatments compared to S1. *Salmonella* was exclusively detected in S1 treatment. *Bacillus*, *Paenibacillus*, and *Microbacterium* were identified as unique genera in S1M treatment, whereas *Lactococcus*, *Serratia*, and *Acinetobacter* were found only in S1MA treatment.

In fluvo-aquic soil, the number of bacterial genera decreased from 20 in S2 to 10 in S2M and 8 in S2MA (Figure 3b). The genera shared across all treatments included *Enterobacter*, *unclassified_f_Enterobacteriaceae*, *Pseudomonas*, *Erwinia*, *Stenotrophomonas*, and *unclassified_o_Enterobacterales*. Among the 13 unique genera in S2 treatment, *Chryseobacterium* and *Lactococcus* exhibited relatively higher abundances. The unique species, *Salmonella* and *Eubacterium_hallii_group* in S2M treatment were potentially derived from manure contamination, while *Glutamicibacter* was a unique genus in S2MA treatment. In contrast, red soil exhibited the lowest diversity of CREB, with unique genera present only in S3 and S3MA treatments (Figure 3c). The genera shared across all red soil treatments included *Enterobacter*, *unclassified_f_Enterobacteriaceae*, and *unclassified_o_Enterobacterales*, while *Erwinia* and *Pseudomonas* were exclusively detected in S3M and S3MA treatments. The S3 treatment contained unique genera such as *Sporosarcina* and *Trichococcus*, whereas *Microbacterium* and *Faecalibacterium* were specifically identified in S3MA treatment. Across pakchoi grown in all three soil types, *Enterobacter* abundance increased in both manure and CTC-manure treatments.

### 3.4. The Functional Prediction of PICRUSt

To further investigate the metabolic potential differences of CREB across treatments under manure application, we performed PICRUSt functional prediction using Kyoto Encyclopedia of Genes and Genomes (KEGG) and Clusters of Orthologous Groups (COG) databases (Figure 4). The Level 1 KEGG pathway analysis revealed that metabolism was the most abundant across all treatments (3.50 × 10^7^–1.32 × 10^8^). Compared to control treatments, the metabolism abundance decreased in manure and CTC-manure treatments (Figure 4a). Environmental information processing decreased by 72.28% and 73.56% in S1MA and S3MA treatments, respectively. Furthermore, human diseases exhibited reduced abundance across all manure and CTC-manure treatments in the three soil types compared to control treatments. Further analysis revealed higher abundance of genetic information processing in S2M and S2MA treatments compared to S1M, S3M, S1MA and S3MA treatments. Organismal systems consistently exhibited the lowest relative abundance across all treatments, with a notable decreasing trend in both manure and CTC-manure amendments, particularly in red soil under CTC-manure treatment (S3MA), which exhibited the most pronounced reduction.

COG functional analysis further elucidated distinct microbial functional profiles across treatments (Figure 4b). Amino acid transport and metabolism had the highest relative abundance among all functional categories, ranging from 7.59% to 9.29%. This function was enriched in the CTC-manure treatments (S1MA: 9.29%, S2MA: 8.73%, S3MA: 10.41%) compared to the control and manure treatments across all soil types. Furthermore, the relative abundance of general function prediction only, energy production and conversion, replication, recombination and repair, cell wall/membrane/envelope biogenesis, inorganic ion transport and metabolism, carbohydrate transport and metabolism, and transcription were relatively high. Energy production and conversion maintained a relatively high level (5.44–7.04%) in all treatments, with the lowest value observed in S3MA (5.44%). For carbohydrate transport and metabolism, it fluctuated greatly among different treatments (4.86–6.67%), with the lowest relative abundance in the S1MA treatment. Transcription showed an increasing trend in both S3MA (7.39%) and S1MA (7.44%) treatments. The relative abundance of secondary metabolites biosynthesis was also relatively high in S1MA and S3MA treatments. In addition, signal transduction mechanisms increased abnormally in S1MA treatment (6.81%), approximately twice that of the other treatments. Furthermore, the relative abundance of function unknown remained consistently high across all treatments, ranging from 9.79% to 10.84%.

### 3.5. Analysis of the Impact Factors of CREB

The structure of CREB communities were correlated with soil total nitrogen (TN), total phosphorus (TP), and organic matter (OM) (Figure 5a). Black soil exhibited higher TN concentrations (1.17–1.30 g/kg) compared to fluvo-aquic soil (0.52–0.66 g/kg) and red soil (0.57–0.63 g/kg) (Appendix A). Although the differences were not statistically significant, TN concentration decreased in all CTC-manure treatments across the three soil types. Manure application generally increased TP concentration, with increases of 2.1% in S1M compared to S1, and 7.5% in S3M compared to S3. OM showed the most pronounced changes in black soil, with significantly lower values in S1M and S1MA treatments compared to S1 (*p* < 0.05) (Appendix A).

Further correlation analysis between CREB in pakchoi and CRB in soil revealed that CREB was mainly influenced by *Pseudomonas*, *Bacillus*, *Stenotrophomonas*, *Microbacterium*, *Chryseobacterium*, *Brevibacillus*, *Paenibacillus*, and *Lysinibacillus* in the soil. Six genera, *Pseudomonas*, *Bacillus*, *Stenotrophomonas*, *Microbacterium*, *Chryseobacterium*, and *Paenibacillus*, were detected in both soil and pakchoi (Figure 2b and Figure 5b). *Brevibacillus* and *Lysinibacillus* were not found in the CREB of pakchoi. Correlation analysis indicated an extremely significant positive correlation between *Brevibacillus* and *Paenibacillus* and *Lysinibacillus* in the soil (*p* < 0.01). Similarly, *Enterobacter* showed an extremely significant positive correlation with *Stenotrophomonas* and *Microbacterium* (*p* < 0.01), and *Ralstonia* was extremely significantly positively correlated with *Bacillus*.

## 4. Discussion

### 4.1. Influence of Manure and CTC on CREB Abundance in Pakchoi

The observed increase in TCEB abundance in manure treatments is consistent with previous findings. A study demonstrated that pakchoi grown in chicken manure soil exhibited increased TCEB levels, with abundances of 0.71 × 10^9^ CFU/g in roots, 3.12 × 10^9^ CFU/g in stems, and 4.03 × 10^9^ CFU/g in leaves [28]. In contrast, the reduction in absolute TCEB abundance in CTC-manure treatments relative to manure-only treatments indicates that CTC exerts inhibitory effects on TCEB populations. CTC is known to exert its antibacterial effects through specific binding to the 30S ribosomal subunit, thereby inhibiting protein synthesis. This mechanism confers significant antimicrobial activity against both Gram-positive bacteria (e.g., *Staphylococcus* and *Streptococcus*) and certain Gram-negative bacteria (e.g., *Pasteurella*) [29]. Furthermore, CTC can translocate into plants through the root system and subsequently suppress or eliminate a portion of endophytic bacteria. Antibiotic residues were detected in all CTC-manure treatments, with concentrations of 4.48 μg/g, 0.87 μg/g, and 1.21 μg/g in S1MA, S2MA, and S3MA, respectively (Appendix A). These results demonstrate that CTC significantly reduces TCEB abundance, exhibiting a stronger suppressive effect than manure application alone.

The presence of CREB across all treatments, including the lower abundance in S1 treatment, aligns with previous studies indicating that fresh vegetables naturally carry certain levels of ARB [30,31]. The marked elevation of CREB abundance in CTC-manure treatments highlights that both manure and CTC-manure collectively promote the proliferation of CREB populations within the plant endosphere. These findings provide empirical evidence that agricultural amendments, including poultry waste and its composites with antibiotics, significantly enhance the prevalence of antibiotic resistance determinants in soil ecosystems, thereby facilitating the colonization of vegetable endophytes by ARB [32]. The differential distribution patterns of CREB across soil types and amendment strategies further indicate that the proliferation of ARB within the pakchoi endophytic system is not only influenced by manure and antibiotic inputs but is also significantly modulated by soil type. The most pronounced effects of CTC-manure on CREB relative abundance in black and fluvo-aquic soils further underscore the importance of soil properties in the dissemination and colonization of antibiotic resistance in vegetable endophytes.

### 4.2. The Influence of Manure and CTC on CREB Community Structure

The consistent presence of Proteobacteria at significant levels across all treatments (Figure 2a) suggests that this phylum may inherently represent a dominant group of indigenous endophytes in plants. A previous study has demonstrated that soil-derived ARB can enter the plant endophytic system through root uptake [21]. Proteobacteria are also the main ARB in the soil in our previous study [12]. These results demonstrate that Proteobacteria may inherently represent a dominant phylum of indigenous endophytes in plants. Additionally, their prevalence could also result from the translocation of soil-derived Proteobacteria into the endophytic system of pakchoi via root uptake.

The consistent detection of *unclassified_f_Enterobacteriaceae* and *Enterobacter* across all treatments suggests that these two genera represent widespread ARB in plants. *Acinetobacter*, which dominated in S1MA treatment, is a genus of aerobic, Gram-negative bacteria widely distributed in soil and aquatic environments [33]. Extensive research has demonstrated its multidrug-resistant properties against various antibiotics [34]. Notably, *Acinetobacter*, a clinically significant representative of this genus, is a major causative agent of pneumonia, capable of surviving for weeks under dry conditions while maintaining high transmissibility [35]. The significant positive correlation between *Serratia* and *Acinetobacter* observed in this study (Appendix A), indicates that CTC exposure significantly promoted their proliferation, concurrently elevating the risk of pathogenic bacterial transmission in pakchoi. A marked increase in Stenotrophomonas was observed in the S2M treatment. This finding aligns with a recent soil microbiome study that the detection of this genus in agricultural soils [12]. This supports the possibility of ARB transfer from soil to plants. In contrast, *Streptomyces* maintained considerable abundance in soil but was undetectable in the plant endophytic system. Similarly, *Ralstonia* was identified as a predominant CRB in red soil but was not detected in the CREB of pakchoi. The increased relative abundance of *Pseudomonas* in CTC-manure treatments across all soil types indicates its strong antibiotic resistance and suggests that the presence of antibiotics induced its growth and proliferation. *Pseudomonas* is a common bacterial group in the environment, with *Pseudomonas aeruginosa* being the most notable genera. As an obligate aerobe, it is an opportunistic pathogen capable of infecting various animals and a frequent cause of nosocomial infections. It can survive for extended periods in purified water and readily forms biofilms, with multidrug resistance making it a major challenge in clinical settings [36,37]. Its detection in soil also implies potential soil–plant transmission routes.

Notably, *Pseudomonas* exhibited a significant positive correlation with *Erwinia* in pakchoi grown in fluvo-aquic soil, but a negative correlation in red soil (Appendix A). These findings demonstrate that soil type variations fundamentally alter their co-occurrence relationships and ecological interactions. *Erwinia*, a genus of Gram-negative, rod-shaped, facultatively anaerobic bacteria within the *Enterobacteriaceae* family, is phylogenetically close to *Escherichia*. These bacteria are ubiquitously distributed in natural environments, including soil, water systems, and plant surfaces. Several *Erwinia* genera are important plant pathogens, causing substantial economic losses in agriculture [38,39]. The exclusive increase in Erwinia abundance in S3MA treatment indicates that the red soil environment provides favorable conditions for its proliferation. The absence of *Streptomyces* in plant endophytic systems, despite its considerable abundance in soils, suggests its ecological niche may be restricted to soil environments. *Streptomyces* represents a group of Gram-positive, aerobic Actinobacteria that are widely distributed in soils, decaying organic matter, and marine ecosystems [40,41]. Similarly, *Ralstonia* was not detected in pakchoi CREB, even though it was predominant in the red soil. This indicates that it might be exclusively soil-adapted and incapable of transferring to plants.

*Rhodococcus* was detected in all black soil treatments, indicating that this characteristic soil bacterium is well-adapted to black soil and may be taken up by plant roots. The detection of *Faecalibacterium* in red soil is notable, as this Gram-positive Firmicutes bacterium is typically found in intestinal environments [42]. Its presence in S3MA treatment suggested that CTC selection pressure might facilitate the dissemination of such intestinal bacteria in this particular soil environment. As Gram-negative, facultative anaerobic bacteria, *Enterobacter* are widely distributed in soils, plant rhizospheres, and endophytic compartments. While some strains act as opportunistic human pathogens, numerous plant-associated *Enterobacter* strains have demonstrated significant plant growth-promoting functions. These beneficial effects are mediated through multiple mechanisms including nitrogen fixation, phosphate solubilization, and phytohormone production, ultimately enhancing plant growth and stress tolerance in agricultural ecosystems [43,44]. This suggests a dual origin for the detected *Enterobacter*. One portion may have originated from the plant’s endogenous microbiome. Another portion could have entered via soil–root transmission pathways.

### 4.3. The Influence of Manure and CTC on the Functional Prediction of PICRUSt

The observed reduction in metabolism abundance in manure and CTC-manure treatments may result from either manure-induced alteration in soil microenvironments, or antibiotic stress inhibiting key microbial metabolic enzymes or reducing functional microbial populations. A previous study has demonstrated distinct metabolic functional profiles in microbial communities exposed to different antibiotics compared to antibiotic-free conditions [45]. The reduction in environmental information processing might correlate with the significantly lower TCEB abundance observed in these treatments (Figure 1a). Furthermore, the abundance of human disease-associated ARGs decreased across all amended treatments, suggesting that both manure application and CTC exposure may reduce their enrichment. Manure application may reduce ARG transmission risks by stimulating beneficial microorganisms (decomposers and nitrogen-fixing bacteria), which competitively suppress bacteria harboring human pathogen-associated ARGs. In contrast, CTC appears to selectively inhibit pathogens carrying clinically relevant ARGs (*tet* genes) while concurrently enriching environmentally adaptive resistance mechanisms in indigenous soil microbiota. For instance, *Pseudomonas* strains function as typical plant growth-promoting rhizobacteria that suppress phytopathogens through antibiotic production, nutrient competition, and induced systemic resistance [46]. In this study, *Pseudomonas* abundance showed increasing trends in both manure and CTC-manure treatments (Figure 2b), which may partly explain the observed reduction in human disease-related functions.

The higher abundance of genetic information processing in S2M and S2MA treatments suggests that CREB in fluvo-aquic soil may have an enhanced capacity for HGT. The consistently decreasing trend in organismal systems indicates that exogenous inputs (manure and antibiotics) may disrupt host–microbe interactions in soil microbial communities. A previous study indicated that antibiotics remain more stable under acidic conditions, potentially prolonging their inhibitory effects on microbial functions [47]. Soil physical and chemical properties analysis further revealed that manure application exerted more pronounced effects on red soil properties compared to black and fluvo-aquic soils (Appendix A). These alterations likely suppressed microorganisms dependent on host interactions through modified soil conditions. The elevation in amino acid transport and metabolism in CTC-manure treatments may reflect microbial adaptive responses to CTC stress, where enhanced amino acid metabolism serves to sustain essential cellular functions under antibiotic pressure [48]. This is consistent with the lowest abundance of organismal systems in the S3MA treatment (Figure 4a). The difference in carbohydrate transport and metabolism among treatments may reflect the adaptive changes in carbon source utilization strategies among different soil types under CTC selection pressure. An increasing trend in transcription was observed in the S1MA and S3MA treatments. This finding is consistent with previous reports that antibiotic stress can activate microbial transcriptional regulation [45,49]. The abnormal increase in signal transduction mechanisms in S1MA treatment indicated that CTC had a greater impact on the cellular signal transduction system of endophytic microorganisms in pakchoi grown in black soil than in the other two types of soil. Overall, CTC has the greatest impact on the functions of pakchoi grown in black soil and red soil. These differences may stem from the regulatory effects of different soil types on the bioavailability of CTC. The acidic environment of red soil may enhance the activity of CTC, whereas the high organic matter content in black soil may reduce its bioavailability through adsorption.

### 4.4. The Influence of Soil Physical and Chemical Properties and CRB on CREB of Pakchoi

The observed decreases in TN across CTC-manure treatments suggest that CTC may inhibit the activity of nitrogen-transforming microorganisms. This aligns with established ecological strategies of soil microbes in response to nitrogen availability. Specifically, under reduced TN conditions, oligotrophic microbes likely gain a competitive advantage by upregulating ammonium assimilation genes, while copiotrophic populations become suppressed due to limited nutrients [50]. The increase in TP following manure application likely promoted the proliferation of phosphate-solubilizing bacteria such as *Pseudomonas* [13], which is consistent with its increased abundance observed in both manure and CTC-manure treatments in this study (Figure 2b). As an essential element for nucleic acid synthesis, increased TP content may facilitate the HGT frequency of ARGs by providing more abundant substrates for genetic material synthesis, potentially contributing to the observed CREB abundance increase across treatments. The significant reduction in OM in black soil under manure application may be explained by previous research. It demonstrated that manure application, as a labile carbon input, significantly activates native soil microbial activity and accelerates the decomposition of indigenous OM [51].

These mechanistic insights are consistent with the shifts in microbial metabolic profiles observed after manure and CTC-manure application (Figure 4). The detection of six bacterial genera common to both soil and pakchoi indicates that a portion of CREB in pakchoi originates from soil and enters the plant via the root system. Additionally, resistance developed by the endophytic bacteria within the plant itself contributes to CREB composition. Studies have shown that organic fertilizer application promotes the rhizosphere environment and increases the abundance of ARGs in the underground parts of plants. Exogenous substances taken up by roots can be transported to aerial parts through transpiration, which may serve as a key driver for the internal movement of free ARB within plants [52,53]. ARB and ARGs typically demonstrate higher diversity and abundance in rhizosphere endophytes compared to phyllosphere communities, likely due to progressive resistance during upward ARB migration leading to reduced transmission efficiency. This study’s soil–plant comparative analysis further confirmed that soil CRB communities exhibited significantly greater abundance and diversity than plant-associated CREB [12]. The absence of *Brevibacillus* and *Lysinibacillus* in pakchoi CREB, along with their strong correlations with other soil bacteria, suggests that these genera indirectly affect the CREB community structure in plants by influencing indigenous microorganisms in the soil. These findings collectively indicate the existence of a complex interaction network between soil CRB and plant CREB, which leads to differences in CREB abundance and communities under different soil types and treatment conditions.

## 5. Conclusions

This study analyzed pakchoi grown in different soil types and discovered significantly increased abundance of CREB in edible portions (1.71 × 10^3^–3.61 × 10^3^ CFU/g) under the CTC-manure treatment, with relative abundances following the order S1MA > S2MA > S3MA. The predominant CREB genera were *unclassified_f__Enterobacteriaceae*, *Pseudomonas* and *Enterobacter*. Elevated abundance of pathogenic bacteria such as *Pseudarthrobacter*, *Klebsiella*, *Corynebacterium* and *Rhodococcus* was also observed with manure or CTC-manure treatments. Distinct unique species were demonstrated across treatments through network analysis. *Serratia* and *Acinetobacter* were found in S1MA, *Glutamicibacter* dominated in S2MA, while *Microbacterium* and *Faecalibacterium* were enriched in S3MA. Furthermore, significant metabolic disturbances were exhibited by CTC in the red soil system (S3MA), including inhibited energy metabolism and enhanced amino acid metabolism, whereas activation of signal transduction systems and promotion of secondary metabolite accumulation primarily occurred in the black soil (S1MA). These findings provide a scientific basis for incorporating antibiotic resistance and pathogen risk assessment into the safety management of agricultural manure.

## Figures and Tables

**Figure 1 microorganisms-13-02398-f001:**
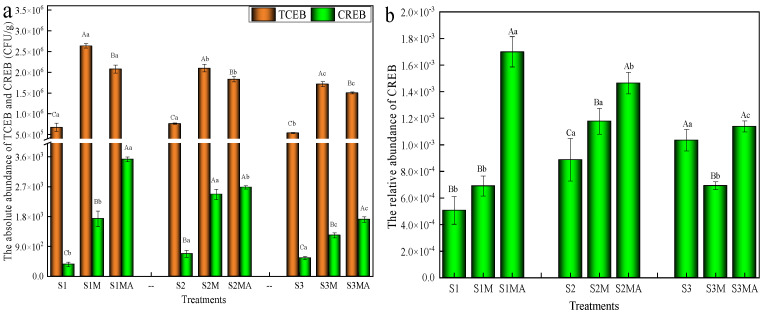
The absolute and relative abundances of TCEB and CREB in endogenous system of pakchoi. The absolute abundance of TCEB and CREB (**a**). The relative abundance of CREB (**b**). The upper-case letters represent the differences between different treatments of the same soil types, while lower-case letters represent the differences between the same treatments of different soil types.

**Figure 2 microorganisms-13-02398-f002:**
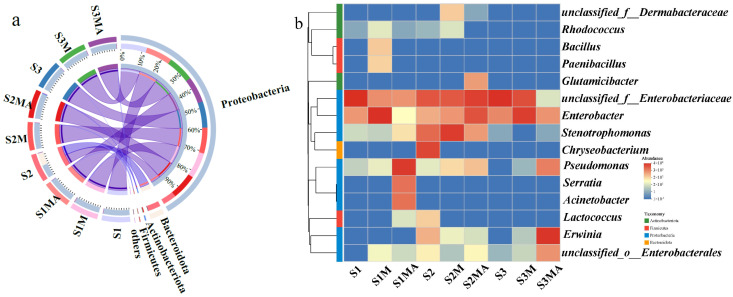
Changes in abundance and community structure of CREB in the phylum (**a**) and genus (**b**) levels.

**Figure 3 microorganisms-13-02398-f003:**
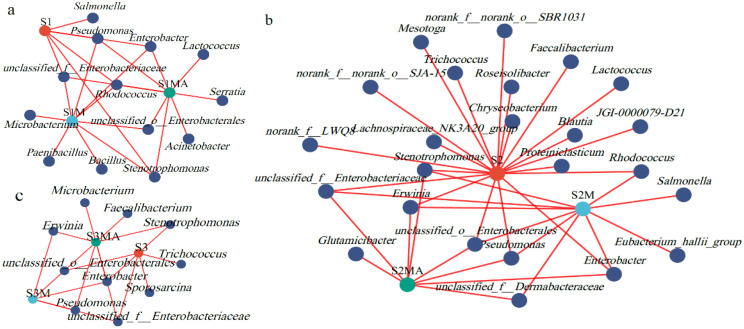
Analysis of the differences in CREB communities. CREB of pakchoi grown in black soil (**a**). Grown in fluvo-aquic soil (**b**). Grown in red soil (**c**).

**Figure 4 microorganisms-13-02398-f004:**
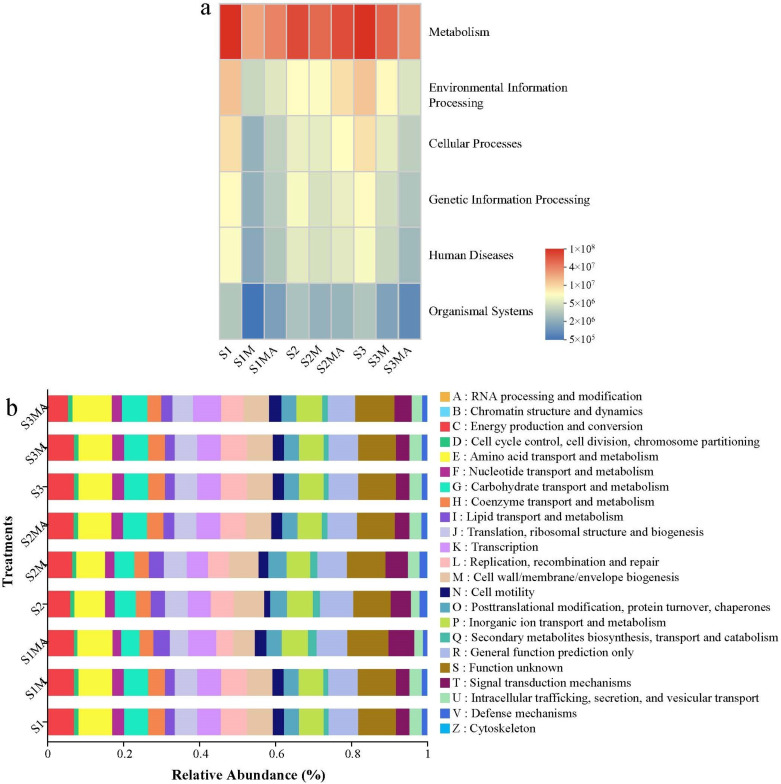
Abundances of prediction pathways of level 1 KEGG (**a**) and COG (**b**).

**Figure 5 microorganisms-13-02398-f005:**
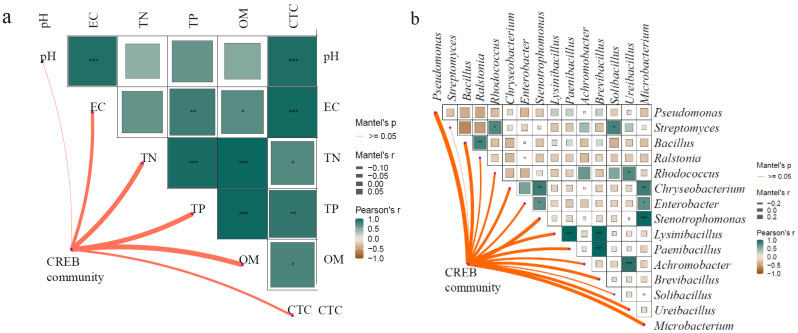
The correlation analysis of CREB and soil physical and chemical properties (**a**) and soil CRB (**b**). * indicates 0.01 < *p* ≤ 0.05; ** indicates 0.001 < *p* ≤ 0.01; *** *p* ≤ 0.001.

## Data Availability

The original contributions presented in the study are included in the article/Appendix A. Further inquiries can be directed to the corresponding author.

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
