# Peer review of "Migration Characteristics of Manure-Derived Antibiotic-Resistant Bacteria in Vegetables Under Different Soil Types"

_microorganisms, 2025, doi:10.3390/microorganisms13102398_

Round 1
Reviewer 1 Report
Comments and Suggestions for Authors
This is a topic of utmost importance. The truth is that there is currently a push to use organic inputs to promote food production. However, it is often overlooked that, if these inputs are not used properly, they can promote the spread of antibiotic-resistant microorganisms.
In general, the introduction requires improvement in both wording and content. Specifically, it should provide details on the study units involved in the research, including soil type and pak choi, the crop on which the research will focus.
The introduction is good, but it sometimes shifts from general to specific and then revisits general ideas. For example, "Plants harbor a substantial microbial community (termed the plant microbiome), which serves as a critical determinant of plant health, fitness, and productivity." This causes the reader to lose track of consecutive ideas. I suggest refining the wording to move from general to specific, thereby guiding the reader more effectively.
Within the introduction, there are also some repetitive ideas, for example, "Although extensive research has been conducted on the diversity and abundance of soil ARB, studies investigating their subsequent transfer into plant microbiomes remain limited," and "However, the current understanding of antibiotic resistance related to plants remains limited compared to soil and wastewater systems."
In the introduction, mention if there is evidence that resistant microorganisms have been transmitted through foods treated with organic fertilizers.
At the end of the introduction, the effects of soil types and antibiotic-resistant bacteria are mentioned. However, the introduction does not define what soil types are, nor does it define whether there is any research on this.
On the other hand, within the introduction, the abundance of ARB in pak choi is mentioned, and it would be worth including the scientific name of this crop because there are countries where we do not consume it. It would be easier to identify it by its scientific name. Furthermore, what is the importance of this crop in terms of production and consumption?
In the methodology, describe the reasons for choosing the different soil types and what was unique about them. Also mention where they were obtained and the characteristics of the sites, as this may correlate with the associated microbiota.
Why use Usearch instead of Qiime2 for high-throughput sequencing analysis?
In the results, did they fail to identify culturable microorganisms?
Figure 2b is of poor quality and needs to be improved.
What we are really seeing in Figures 2 and 3 are antibiotic-resistant bacteria.
How do the microorganisms identified with the high-throughput sequences relate to the culturable ones?
The results describe the physicochemical characteristics of the soil, but the methodology does not describe how these studies were conducted.
In Figure 5b, spell the scientific names correctly.
In general, the discussion is good. It is long, but it explains what is observed in the results.
Moreover, in the conclusion, it would be helpful to add the authors' perspective on what would follow from the results of their work.
Author Response
Comment 1: This is a topic of utmost importance. The truth is that there is currently a push to use organic inputs to promote food production. However, it is often overlooked that, if these inputs are not used properly, they can promote the spread of antibiotic-resistant microorganisms.
Response 1: We are grateful to the reviewer for this valuable comment, which underscores the core motivation of our work. We concur completely that while organic inputs are vital for sustainable agriculture, their potential to disseminate antibiotic-resistant microorganisms is a significant and often underestimated risk. Our study was designed explicitly to address this issue. Through a systematic examination of manure and CTC-manure applications across different soil types, we have documented how these practices can indeed alter the abundance and community structure of antibiotic-resistant bacteria in the soil-plant system. The broader implications of our findings for agricultural practice are elaborated in the Results and Discussion.
Comment 2: In general, the introduction requires improvement in both wording and content. Specifically, it should provide details on the study units involved in the research, including soil type and pakchoi, the crop on which the research will focus.
Response 2: We are extremely grateful for your valuable comments. We have thoroughly revised the introduction to improve the overall wording for better clarity and flow. More importantly, we have significantly expanded the content to provide essential background on the study units, as suggested. We have specified the type of soil used in this study (this study investigates the abundance variations of ARB in the endophytic systems of pakchoi grown in different soil types (black, fluvo-aquic and red soils) under various manure treatments). Additonally, we highlighted why pakchoi serves as an ideal model crop for this study, especially given its high consumption and potential significance in the human exposure pathway to soil contaminants (Pakchoi (Brassica chinensis) is a staple leafy vegetable in many parts of the world, with particularly high cultivation and consumption levels throughout East Asia. It is safety directly impacts a significant portion of the vegetable supply chain). These additions have strengthened the foundation of our research and more clearly defined the scope and relevance of our work. We believe the introduction is now substantially improved.
Comment 3: The introduction is good, but it sometimes shifts from general to specific and then revisits general ideas. For example, "Plants harbor a substantial microbial community (termed the plant microbiome), which serves as a critical determinant of plant health, fitness, and productivity." This causes the reader to lose track of consecutive ideas. I suggest refining the wording to move from general to specific, thereby guiding the reader more effectively.
Response 3: Thank you very much for your professional comments. We have revised the text to ensure a smoother transition from general concepts to specific research gaps. Additionally, we have thoroughly re-examined the entire introduction to ensure a clear, consecutive narrative that moves from broad concepts to the specific focus of our study. This change eliminated the previously disjointed structure and guides the reader more effectively.
Revised in the text: Plants harbor a substantial microbial community, known as the plant microbiome, which plays a critical role in plant health, fitness, and productivity [14, 15]. Extensive research has focused on the diversity and abundance of ARB in soil. However, studies examining their subsequent transfer into the plant microbiome remain limited [11, 14, 16].
Comment 4: Within the introduction, there are also some repetitive ideas, for example, "Although extensive research has been conducted on the diversity and abundance of soil ARB, studies investigating their subsequent transfer into plant microbiomes remain limited," and "However, the current understanding of antibiotic resistance related to plants remains limited compared to soil and wastewater systems."
Response 4: We sincerely appreciate your valuable comments. We agree that these sentences conveyed overlapping ideas. We have carefully examined the entire introduction, consolidated the repetitive concepts, and refined the text to improve conciseness and logical flow. The revised version has presented a more streamlined and focused argument.
Comment 5: In the introduction, mention if there is evidence that resistant microorganisms have been transmitted through foods treated with organic fertilizers.
Response 5: Thank you for your helpful review. We apologize if this point was not sufficiently prominent in our original manuscript. We did include a discussion on the transmission of ARB via the food chain in the Introduction, supported by references [19, 20, 21, 22, 25]. We have further emphasized this point in the revised text to ensure it is clearly presented.
References: 19. Xu, H.; Chen, Z.Y.; Huang, R.Y.; Cui, Y.X.; Li, Q.; Zhao, Y.H.; Wang, X.L.; Mao, D.Q.; Luo, Y.; Ren, H.Q. Antibiotic Resistance Gene-Carrying Plasmid Spreads into the Plant Endophytic Bacteria Using Soil Bacteria as Carriers. Environ. Sci. Technol. 2021, 55, 10462–10470. https://doi.org/10.1021/acs.est.1c01615.
- Yang, C.W.; Hsiao, W.C.; Chang, B.V. Biodegradation of Sulfonamide Antibiotics in Sludge. Chemosphere 2016, 150, 559–565. https://doi.org/10.1016/j.chemosphere.2016.02.064.
- Cerqueira, F.; Matamoros, V.; Bayona, J.M.; Berendonk, T.U.; Elsinga, G.; Hornstra, L.M.; Piña, B. Antibiotic Resistance Gene Distribution in Agricultural Fields and Crops. A Soil-to-Food Analysis. Environ. Res. 2019, 177, 108608. https://doi.org/10.1016/j.envres.2019.108608.
- Chen, M.; Qiu, T.; Sun, Y.; Song, Y.; Wang, X.; Gao, M. Diversity of Tetracycline- and Erythromycin-Resistant Bacteria in Aerosols and Manures from Four Types of Animal Farms in China. Environ. Sci. Pollut. Res. 2019, 26, 24213–24222. https://doi.org/10.1007/s11356-019-05672-3.
- Wei, H.; Ding, S.; Qiao, Z.; Su, Y.; Xie, B. Insights into Factors Driving the Transmission of Antibiotic Resistance from Sludge Compost-Amended Soil to Vegetables under Cadmium Stress. Sci. Total Environ. 2020, 729, 138990. https://doi.org/10.1016/j.scitotenv.2020.138990.
Comment 6: At the end of the introduction, the effects of soil types and antibiotic-resistant bacteria are mentioned. However, the introduction does not define what soil types are, nor does it define whether there is any research on this.
Response 6: We appreciate the reviewer’s constructive comments. In the revised manuscript, we have added the definition of the specific soil types used in this study. Furthermore, we have explicitly stated that there are currently no studies exploring the impact of these soil types on endogenous ARB in plants, which underscores the novelty and necessity of our current research.
Revised: Therefore, this study investigates the abundance variations of ARB in the endophytic systems of pakchoi grown in different soil types (black, fluvo-aquic and red soils) un-der various manure treatments.
However, current research on the effects of soil types on endophytic ARB in plants remains scarce.
Comment 7: On the other hand, within the introduction, the abundance of ARB in pak choi is mentioned, and it would be worth including the scientific name of this crop because there are countries where we do not consume it. It would be easier to identify it by its scientific name. Furthermore, what is the importance of this crop in terms of production and consumption?
Response 7: Thank you very much for your professional comments. We agree that including the scientific name and contextualizing the crop's importance enhances the clarity and global relevance of our introduction. Accordingly, we have added the scientific name Brassica chinensis upon the mention of pakchoi. Furthermore, we have included the sentence to highlight its agronomic and economic significance.
Revised in the text: Pakchoi (Brassica chinensis) is a staple leafy vegetable in many parts of the world, with particularly high cultivation and consumption levels throughout East Asia. It is safety directly impacts a significant portion of the vegetable supply chain [26].
References: 26. Liu, D. L.; Zeleke, K. T.; Wang, B.; Macadam, I.; Scott, F.; Martin, R. J. Crop Residue Incorporation Can Mitigate Negative Climate Change Impacts on Crop Yield and Improve Water Use Efficiency in a Semiarid Environment. European Journal of Agronomy 2017, 85, 51–68. https://doi.org/10.1016/j.eja.2017.02.004.
Comment 8: In the methodology, describe the reasons for choosing the different soil types and what was unique about them. Also mention where they were obtained and the characteristics of the sites, as this may correlate with the associated microbiota.
Response 8: Thanks very much for your helpful comment. The detailed descriptions of the three soil types, including collection sites, and the basis physical and chemical properties were provided in our previous published work. We also made references in 2.1. Experimental Design (The specific experimental design is the same as that of a previous study [12]). To avoid unnecessary repetition and for a detailed description of plant research methods, we choose the soil section to refer to previous articles.
Reference: 12. Song, T.T.; Sardar, M.F.; Wang, X.R.; Li, B.X.; Zhang, Z.Y.; Wu, D.M.; Zhu, C.X.; Li, H.N. Distribution of Antibiotic Resistant Bacteria in Different Soil Types Following Manure Application. Soil Ecol. Lett. 2024, 6, 230210. https://doi.org/10.1007/s42832-023-0210-6.
Comment 9: Why use Usearch instead of Qiime2 for high-throughput sequencing analysis?
Response 9: Thank you very much for your professional comments. We selected Usearch for three primary reasons. First, its algorithm demonstrates strong rigor in OTU clustering by integrating denoising, chimera filtering, and clustering processes, effectively minimizing spurious OTUs. Second, it ensures high-quality analytical outputs while maintaining consistency with our laboratory's established protocols and previous publications. Third, our research team possesses extensive validated experience with this pipeline. Together, these factors guarantee the reliability and comparability of our research outcomes.
Comment 10: In the results, did they fail to identify culturable microorganisms?
Response 10: Thanks for your comments. We successfully isolated and identified culturable microorganisms from all samples. Section 3.1 and Figure 1 present the quantitative results, including both absolute and relative abundances of total cultivable endophytic bacteria (TCEB) and chlorotetracycline-resistant endophytic bacteria (CREB) in the pakchoi endophytic system. Furthermore, Section 3.2 and Figure 2 detail the community composition and structure of CREB, providing comprehensive characterization of the cultured isolates.
Comment 11: Figure 2b is of poor quality and needs to be improved.
Response 11: Thank you very much for your kind comments. We have regenerated the figure at a higher resolution and significantly enhanced its visual clarity by enlarging all text labels.
Comment 12: What we are really seeing in Figures 2 and 3 are antibiotic-resistant bacteria.
Response 12: Thank you very much for your careful comments. Actually, all three figures specifically characterized antibiotic-resistant bacteria. The quantitative analysis of antibiotic-resistant bacteria obtained through plate counting has been presented in Figure 1. The taxonomic composition of these resistant bacteria has been systematically resolved in Figure 2, with the phylum-level distribution shown in Figure 2a and the genus-level profile detailed in Figure 2b. Furthermore, the comparative analysis of ARB community structures across treatments, highlighting shared and unique resistant species through network diagrams, has been provided in Figure 3.
Comment 13: How do the microorganisms identified with the high-throughput sequences relate to the culturable ones?
Response 13: Thank you very much for your professional comments. We would like to clarify that the microorganisms identified through high-throughput sequencing in this study were directly derived from the same pool of colonies grown on our culture plates. To maintain the original proportion of bacterial groups and avoid introducing additional culture bias, we deliberately omitted the conventional liquid culture enrichment step. Instead, we collectively harvested all colonies obtained from CTC-supplemented LB agar plates after the 24-hour incubation, and processed them directly through centrifugation and DNA extraction for sequencing analysis.
This methodological approach ensures that our high-throughput sequencing data represents the comprehensive genetic profile of the entire culturable bacterial community that grew under our specific experimental conditions. While the plate counting provided quantitative data on chlortetracycline-resistant endophytic bacteria (CREB), the subsequent sequencing analysis revealed the detailed taxonomic composition and community structure of these same cultivable bacteria. This integrated approach allows us to not only quantify the antibiotic-resistant bacteria but also understand their specific identities within the cultivable fraction of the pakchoi endophytic microbiome, providing complementary insights through these two analytical methods.
Comment 14: The results describe the physicochemical characteristics of the soil, but the methodology does not describe how these studies were conducted.
Response 14: Thank you for your meticulous review. The analysis of the soil's physical and chemical properties was not repeated in this study. These specific soils had been fully characterized in our previous work. We referenced the prior study to avoid unnecessary duplication and for the sake of conciseness. We acknowledged that this connection was not clearly stated originally. Accordingly, we have revised the 'Materials and Methods' section (2.1). The revision explicitly states the basic physicochemical properties of the soils and provides a clear citation to our previous paper. We believe this clarification addresses the reviewer's concern regarding methodological transparency.
Revised in the text: The physical and chemical properties and CTC of the soils were characterized in our previous study [12].
Reference: 12. Song, T.T.; Sardar, M.F.; Wang, X.R.; Li, B.X.; Zhang, Z.Y.; Wu, D.M.; Zhu, C.X.; Li, H.N. Distribution of Antibiotic Resistant Bacteria in Different Soil Types Following Manure Application. Soil Ecol. Lett. 2024, 6, 230210. https://doi.org/10.1007/s42832-023-0210-6.
Comment 15: In Figure 5b, spell the scientific names correctly.
Response 15: Thank you very much for your careful comments. We have corrected this in the revised manuscript. The scientific names in Figure 5b are now spelled correctly and formatted in italics.
Comment 16: In general, the discussion is good. It is long, but it explains what is observed in the results.
Response 16: We thank the reviewer for the positive comments on our discussion. We are pleased that it clearly explains the results. We have also taken the comment on its length into consideration and have streamlined the text where possible to improve conciseness.
Comment 17: Moreover, in the conclusion, it would be helpful to add the authors' perspective on what would follow from the results of their work.
Response 17: Thank you very much for your helpful comments. We agreed that adding our perspective on the implications of our work would strengthen the conclusion. Accordingly, we revised the concluding paragraph to include our viewpoint on the necessary future steps stemming from our findings.
Revised in the text: These findings provide a scientific basis for incorporating antibiotic resistance and pathogen risk assessment into the safety management of agricultural manure.

Reviewer 2 Report
Comments and Suggestions for Authors
The reviewed manuscript, microorganisms-3880968, is dedicated to the experimental study of the abundance variations of antibiotic-resistant bacteria in the endophytic systems of pakchoi grown in different soil types under various manure treatments.
A definite strength of this study is the analysis of different soil types, which broadens the scope of understanding of various soil microbiomes. However, this approach also presents certain challenges. Comparing microbiomes of different soil types based on absolute microorganism counts is not entirely appropriate, as soils differ in composition and thus can support the growth of different microbial groups affecting each other in different ways. Variations in the concentrations of elements essential for microorganisms may explain differences in the number of growing microbes as well as the dominance of different bacterial groups. Furthermore, when assessing the effects of manure and chlortetracycline-containing manure, it is important to consider the microbiome contributed directly by manure itself. Therefore, the results should be interpreted with consideration of the introduced microbiome.
The manuscript also contains several typos and technical errors, which are listed below:
1. Define all abbreviations at their first appearance in the abstract, main text, and in the first figure or table. For example, “UPVC” line 101, “COG” line 260, “KEGG” line 293, etc.
2. Please zoom in on all figures to make the text readable.
3. Check text for typing mistakes. For example, lines 260, 273, 296, etc.
Author Response
Comment 1: The reviewed manuscript, microorganisms-3880968, is dedicated to the experimental study of the abundance variations of antibiotic-resistant bacteria in the endophytic systems of pakchoi grown in different soil types under various manure treatments.
Response 1: We are very grateful to the reviewers for their positive feedback and valuable comments. We particularly thank the reviewer for this accurate summary of our manuscript's core objective. The specific suggestions provided below have been addressed point-by-point in the following responses, and corresponding revisions have been incorporated to improve the quality of our study on the abundance and community variations of antibiotic-resistant bacteria.
Comment 2: A definite strength of this study is the analysis of different soil types, which broadens the scope of understanding of various soil microbiomes. However, this approach also presents certain challenges. Comparing microbiomes of different soil types based on absolute microorganism counts is not entirely appropriate, as soils differ in composition and thus can support the growth of different microbial groups affecting each other in different ways. Variations in the concentrations of elements essential for microorganisms may explain differences in the number of growing microbes as well as the dominance of different bacterial groups. Furthermore, when assessing the effects of manure and chlortetracycline-containing manure, it is important to consider the microbiome contributed directly by manure itself. Therefore, the results should be interpreted with consideration of the introduced microbiome.
Response 2: We sincerely thank the reviewer for this insightful and constructive comment, which positively acknowledges the strength of our study while also raising critical methodological considerations. We fully agree with these points and have thoroughly revised the manuscript to address them.
In our microbial analysis, we comprehensively evaluated both the absolute and relative abundances of the microbial communities (Figure 2a and b). Furthermore, an original control treatment was included in the experimental design to account for the influence of the native soil microbiome. All analyses of microbial community structure were accordingly based on relative abundance data (Figure 3a and b) to ensure an accurate representation of compositional changes.
Furthermore, our previous results demonstrate that the application of manure primarily induced antibiotic resistance in indigenous soil microorganisms rather than facilitating the colonization of manure-derived ARB [12]. First, dominant CRB genera from the manure (Sporosarcina, unclassified_f_Planococcaceae) were absent or not dominant in the treated soils, indicating their poor adaptability. Second, we observed a significant increase in the relative abundance of native soil genera with known resistance traits (Streptomyces, Bacillus) under CTC-manure treatment. The increased resistance in soil is largely attributable to the selection pressure exerted by antibiotics, which prompted the indigenous microbial community to develop resistance. Therefore, we mainly considered the microorganisms that migrated from the soil to the plants and the subsequent changes in the plant endophytic microbial communities.
Reference: 12. Song, T.T.; Sardar, M.F.; Wang, X.R.; Li, B.X.; Zhang, Z.Y.; Wu, D.M.; Zhu, C.X.; Li, H.N. Distribution of Antibiotic Resistant Bacteria in Different Soil Types Following Manure Application. Soil Ecol. Lett. 2024, 6, 230210. https://doi.org/10.1007/s42832-023-0210-6.
Comment 3: Define all abbreviations at their first appearance in the abstract, main text, and in the first figure or table. For example, “UPVC” line 101, “COG” line 260, “KEGG” line 293, etc.
Response 3: Thank you very much for your meticulous comments. We have carefully checked the manuscript and have now defined all abbreviations at their first occurrence.
Specific examples of the changes made include:
Line 101: UPVC (Unplasticized Polyvinyl Chloride)
Line 260: COG (Clusters of Orthologous Groups)
Line 293: KEGG (Kyoto Encyclopedia of Genes and Genomes)
Line 235: CRB (CTC-resistant bacteria)
Comment 4: Please zoom in on all figures to make the text readable.
Response 4: Thank you very much for your kind comments. We have carefully reviewed all figures and increased the font sizes of all labels, axis text, and legends to ensure clear readability, while also enhancing the resolution of the images in our revised manuscript.
Comment 5: Check text for typing mistakes. For example, lines 260, 273, 296, etc.
Response 5: Thank you very much for your helpful comments. We have thoroughly proofread the manuscript to correct all typing mistakes, including those on the lines you indicated (260, 273, 296). All corrections have been highlighted in red for your convenience.

Reviewer 3 Report
Comments and Suggestions for Authors
This is a very interesting paper about the study of the Migration of Resistant Bacteria in Vegetables under Different Soil Types and with added manure.
It is very well-written and clear, with a wealth of important information. It has an interesting literature review, several important experimental methodologies and a dense discussion of the results.
It is only important to point out the absence of some acronyms' definition, a few sentences that should be revised, and the description of the methodology for the CTC quantification.
Please find more comments in the attached document.

Author Response
Comment 1: This is a very interesting paper about the study of the Migration of Resistant Bacteria in Vegetables under Different Soil Types and with added manure. It is very well-written and clear, with a wealth of important information. It has an interesting literature review, several important experimental methodologies and a dense discussion of the results. It is only important to point out the absence of some acronyms' definition, a few sentences that should be revised, and the description of the methodology for the CTC quantification. Please find more comments in the attached document.
Response 1: We are grateful for your positive assessment of our manuscript and your constructive suggestions. Your encouraging remarks on the paper's clarity and interest are much appreciated. We have thoroughly addressed all the points raised and believe the manuscript has been substantially improved as a result.
Comment 2: What does "indigenous soil ARB" mean in the abstract?
Response 2: Thank you very much for your careful comment. In this study, a significant correlation was observed between specific bacterial genera in the soil and endophytes in plants, based on a combined analysis of CREB and ARB (Figure 5b).
Comment 3: Line 40: what is the difference?
Response 3: Thank you for your meticulous review. We agree that our original phrasing was unclear. We have revised in the revised manuscript.
Revised in the text: Organic fertilizers provide essential macro- and micronutrients, along with abundant organic matter, making them popular choices in organic agriculture and green food production.
Comment 4: Line 108: It would be better to change treatments to assays!? or another terminally. In fact, no treatment is being done. Please see along the text.
Response 4: Thank you for your professional review. The soil treatment procedures were conducted using our established methods, which have been described in detail in our previous publication [12]. For brevity, the section was abbreviated in this manuscript, but the full methodology encompasses the following key aspects:
Unplasticized polyvinyl chloride (UPVC) pipes (25 cm diameter, 45 cm length) were filled with soil. Firstly, 4 cm of quartz sand was spread on the bottom of the UPVC pipe which was then filled with sieved soil. The soil without any substance was used as a control. The soil moisture content was adjusted to 60% of the field water holding capacity. Three replicates were set up for each different treatment. The manure treatment was manure mixed into the 0-20 cm topsoil at the equivalent amount of 225 kg/hm2 of TN. The CTC-manure treatment was CTC dissolved in water at 10 mg/kg (CTC: soil), and then mixed with the soil including manure [12].
Reference: 12. Song, T.T.; Sardar, M.F.; Wang, X.R.; Li, B.X.; Zhang, Z.Y.; Wu, D.M.; Zhu, C.X.; Li, H.N. Distribution of Antibiotic Resistant Bacteria in Different Soil Types Following Manure Application. Soil Ecol. Lett. 2024, 6, 230210. https://doi.org/10.1007/s42832-023-0210-6.
Comment 5: Line 136: ?? performed? Line 339: poultry manure? Line 350-351: Repgrase! As it doesn't have no meaning.
Response 5: Thank you very much for your helpful comments. We have revised these issues in the revised manuscript and conducted a detailed examination of the entire text.
Revised in the text: 1. Colony-forming units (CFUs) were then counted to quantify the total cultivable endophytic bacteria (TCEB) and chlortetracycline-resistant endophytic bacteria (CREB).
- These findings provide empirical evidence that agricultural amendments, including poultry waste and its composites with antibiotics, significantly enhance the prevalence of antibiotic resistance determinants in soil ecosystems, thereby facilitating the colo-nization of vegetable endophytes by ARB.
- The consistent presence of Proteobacteria at significant levels across all treatments (Figure. 2a), suggests that this phylum may inherently represent a dominant group of indigenous endophytes in plants. A previous study has demonstrated that soil-derived ARB can enter the plant endophytic system through root uptake.
Comment 6: The issue regarding punctuation marks is in 2.6.
Response 6: Thank you very much for your careful comments. We have revised in the revised manuscript.
Revised in the text: Data visualization and statistical analyses were performed using the following software tools: column charts were generated using Origin 9.1 (OriginLab, San Diego, CA, USA); one-way analysis of variance (ANOVA) was conducted using IBM SPSS Statistics 23.0 (IBM, Chicago, IL, USA) to determine significant differences among treatments (p < 0.05); phylum level variations in bacterial communities were visualized using Circos-0.67-7 (http://circos.ca/); heatmaps were constructed using the PcoA (Principal Coordinate Analysis) package in R 3.5.2 (https://www.r-project.org/); net-work analysis was performed and visualized using Cytoscape 3.3.0 (https://cytoscape.org/).
Comment 7: The labes in the are very difficult to see.
Response 7: Thank you very much for your meticulous comments. We adjusted the position of the label and enlarged the label in the revised manuscript.
Comment 8: Line 301: How is this explaind? With the addition of organic matter from the manure de OM decreases?
Response 8: Thank you for your professional and insightful review. The contrasting responses of soil organic matter (OM) to manure addition across different soil types clearly demonstrate how soil properties govern the direction and magnitude of the priming effect.
In the black soil, the observed decrease in OM is attributed to a strong positive priming effect. This soil type generally possesses a high native OM content, and the introduction of fresh manure served as a readily available energy source that intensely stimulated the native microbial community. This enhanced microbial activity led to the co-metabolism and accelerated decomposition of the pre-existing, stable soil OM, ultimately resulting in a net loss.
Conversely, we observed a net increase in OM in the fluvo-aquic and red soils. This difference can be explained by the occurrence of minimal or even negative priming in these soils, which are typically lower in native OM and nutrients. Here, the microbial community—initially in a resource-limited state—primarily utilized the easily decomposable manure carbon for its own growth and metabolism, without significantly accelerating the decomposition of the native OM. Furthermore, particularly in the red soil, stabilization mechanisms also contributed to this outcome. The high clay content and abundance of iron/aluminum oxides in this soil type facilitate the rapid stabilization of newly added organic carbon from manure through organo-mineral associations, thereby protecting it from immediate microbial decomposition.
Comment 9: Line 329: Nothing is said in the Methodoloyg about the determination of CTC.
Response 9: Thank you for your careful review. The analysis of the soil's CTC was not repeated in this study. These specific soils had been fully characterized in our previous work. We referenced the prior study to avoid unnecessary duplication and for the sake of conciseness. We acknowledged that this connection was not clearly stated originally. Accordingly, we have revised the 'Materials and Methods' section (2.1). The revision explicitly states the basic physicochemical properties of the soils and provides a clear citation to our previous paper.
Revised in the text: The physical and chemical properties and CTC of the soils were characterized in our previous study [12].
Reference: 12. Song, T.T.; Sardar, M.F.; Wang, X.R.; Li, B.X.; Zhang, Z.Y.; Wu, D.M.; Zhu, C.X.; Li, H.N. Distribution of Antibiotic Resistant Bacteria in Different Soil Types Following Manure Application. Soil Ecol. Lett. 2024, 6, 230210. https://doi.org/10.1007/s42832-023-0210-6.